# Compound-V formations in shorebird flocks

Aaron J Corcoran, Tyson L Hedrick*

University of North Carolina at Chapel Hill, Chapel Hill, United States

**Abstract** Animal groups have emergent properties that result from simple interactions among individuals. However, we know little about why animals adopt different interaction rules because of sparse sampling among species. Here, we identify an interaction rule that holds across single and mixed-species flocks of four migratory shorebird species spanning a seven-fold range of body masses. The rule, aligning with a one-wingspan lateral distance to nearest neighbors in the same horizontal plane, scales linearly with wingspan but is independent of nearest neighbor distance and neighbor species. This rule propagates outward to create a global flock structure that we term the compound-V formation. We propose that this formation represents an intermediary between the cluster flocks of starlings and the simple-V formations of geese and other large migratory birds. We explore multiple hypotheses regarding the benefit of this flock structure and how it differs from structures observed in other flocking species.
DOI: https://doi.org/10.7554/eLife.45071.001

## Introduction

The collective movements of animals—from schooling fish to swarming insects and flocking birds—have long excited intrigue among observers of nature. Collective motion arises as an emergent property of interactions between individuals (reviewed by *Herbert-Read, 2016* and by *Vicsek and Zafeiris, 2012*). Thus, much attention has been placed on identifying local interaction rules (*Ballerini et al., 2008a*; *Herbert-Read et al., 2011*; *Katz et al., 2011*; *Lukeman et al., 2010*) and how those rules affect group structure and movement (*Buhl et al., 2006*; *Hemelrijk and Hildenbrandt, 2012*). However, comparative data across species are still limited, preventing us from testing hypotheses regarding the evolution and diversity of collective movement patterns.

Hundreds of bird species fly in groups, but most quantitative research has focused on starlings (*Attanasi et al., 2014*; *Ballerini et al., 2008b*; *Cavagna et al., 2010*), homing pigeons (*Nagy et al., 2013*; *Nagy et al., 2010*; *Pettit et al., 2015*; *Pettit et al., 2013*; *Usherwood et al., 2011*) and birds that fly in V-formations (*Badgerow and Hainsworth, 1981*; *Cutts and Speakman, 1994*; *Hummel, 1983*; *Lissaman and Shollenberger, 1970*; *Maeng et al., 2013*; *Portugal et al., 2014*; *Weimerskirch et al., 2001*). These data indicate that smaller birds fly in relatively dense cluster flocks that facilitate group cohesion and information transfer (*Attanasi et al., 2014*; *Ballerini et al., 2008a*), whereas larger migratory birds fly in highly structured V formations (also known as line or echelon formations) that provide aerodynamic and energetic benefits (*Lissaman and Shollenberger, 1970*; *Portugal et al., 2014*; *Weimerskirch et al., 2001*). However, descriptive accounts of flock structure over a greater range of species (*Heppner, 1974*; *Piersma et al., 1990*) cover a range of flock types, spanning the extremes of V-formation and large cluster flocks. The species whose flocking behavior have been studied quantitatively differ in many ways that could be important for flocking, including body size, ecology, the frequency of aggregation and its behavioral context. Therefore, on the basis of the available data, it is difficult to conclude what factors cause birds to adopt a specific group formation, or even what factors affect interaction rules, positioning and behavior within flocks.

**\*For correspondence:**
thedrick@bio.unc.edu

**Competing interests:** The authors declare that no competing interests exist.

**eLife digest** Birds often fly in flocks ranging from very structured V-formations to unstructured clusters. Many studies have tried to prove what causes birds to flock and how it benefits them. Flocks, for example, may help birds to avoid predators and to navigate. Flying in a V-shaped formation likely also gives aerodynamic benefits that can make it easier to fly long distances.

Few studies, however, have measured how the positions of birds in a flock relate to things like flying speed or the frequency of wing flaps. This is because it was difficult to take such measurements in large flocks of moving birds. Advances in cameras and computers are now making it easier to track individual birds flying in large flocks. The technology allows scientists to measure how birds position themselves in relation to other birds, or how flock-positioning varies by bird size, species, ecology, and behaviors. Such measurements may help scientists better understand why and how birds flock.

Corcoran and Hedrick now show that four different types of shorebirds position themselves in the same way when flying in a flock. In the experiments, digital cameras recorded video of 18 cluster-like flocks of four different species of birds flying over a bird sanctuary or agricultural fields. The flocks ranged in size from a hundred to a thousand birds. Some flocks had two types of bird. The four types of birds – dunlin, short-billed dowitcher, American avocet, and marbled godwit – live in similar environments but greatly vary in size and fly at different speeds. Corcoran and Hedrick measured individual bird positions using three-dimensional computer reconstructions of the flocks.

Each bird – regardless of size or species – most commonly flew about one wingspan to the side and between a half to one-and-a-half wingspans back from the bird in front of it. Birds flying in simple V-shaped formations follow similar rules. This suggests that birds flying in clusters may also gain aerodynamic benefits.
DOI: https://doi.org/10.7554/eLife.45071.002

We aimed to address these questions by collecting three-dimensional (3D) trajectories of the birds in flocks of four shorebird species that have similar ecologies (all forage in large groups in coastal habitats and migrate long distances) but that cover a seven-fold range of body mass and two-fold range of wingspan. Our study species include dunlin (*Calidris alpina* Linnaeus 1758; 56 g, 0.34 m wingspan), short-billed dowitcher (*Limnodromus griseus* Gmelin 1789; 110 g, 0.52 m wingspan), American avocet (*Recurvirostra americana* Gmelin 1789; 312 g, 0.72 m wingspan), and marbled godwit (*Limosa fedoa*, Linnaeus, 1758; 370 g, 0.78 m). Molecular dating indicates that these species diverged from their nearest common ancestor approximately 50 million years ago (Mya) (*Baker et al., 2007*), providing time for evolutionary diversification of flocking behavior. By comparing the group structure of birds across a range of body sizes and by comparing our data with those in the literature, we aimed to determine the extent to which flock structure varies across species with different body sizes and ecologies. We employ three approaches: (1) identification of local interaction rules by quantifying the relative positions of birds and their nearest neighbors; (2) quantification of the degree of spatial structure within flocks; and (3) measurement of individual speeds and wingbeat frequencies to examine how local and global position within the flock affect flights behavior.

On the basis of existing flock data, we hypothesized that flocks of larger shorebird species would be more structured than those of smaller species (recapitulating the trend of larger birds flying in highly structured V formations) and that larger species would also exhibit aerodynamic formations more frequently. Because a previous study showed that flying in a cluster flock is energetically costly in pigeons (*Usherwood et al., 2011*), we hypothesized that birds flying in the middle and rear of flocks and birds flying closer to their nearest neighbor would have reduced flight performance (lower speed relative to their wingbeat frequency). Surprisingly, we found that all four species studied here flew in a flock structure that we term the compound-V formation. We propose that this structure might be an adaptation for aerodynamic flocking in migratory species, and that ecology is an underappreciated driver of the evolution of avian flocking behavior.

## Results

We reconstructed the 3D trajectories from 18 bird flocks that ranged in size from 189 to 1039 individuals, which were recorded for 2.4–13.2 s at 29.97 frames per second (*Figure 1*, *Table 1*). This resulted in 1,598,169 3D position measurements that were used to examine flock structure. Sixteen of the 18 flocks were comprised entirely of a single species. The remaining two flocks were mixed-species flocks of marbled godwits and short-billed dowitchers. Computer vision techniques allowed the species of individuals in mixed-species flocks to be identified on the basis of differences in body size (see 'Materials and methods').

### Nearest-neighbor alignment

We examined flock structure by quantifying the position of each bird with respect to its nearest neighbor. We used modal values to characterize typical neighbor positions because position distributions were skewed as a result of values being cropped at zero. In all flocks, nearest neighbors flying within the same horizontal plane [an elevation slice of ±1 wingspan, a mean of 56% of nearest neighbors across all flocks (range 35–76%)] exhibited a distinctly peaked distribution, where modal neighbor position was offset both in front-back and lateral distance (*Figure 2a,b*). By contrast, nearest neighbor birds flying outside the horizontal elevation slice of ±1 wingspan were distributed randomly with a peak directly above or below the focal bird (*Figure 2c,d*). This indicates that shorebirds adopt alignment rules for neighbors flying within their same elevation slice. On average, birds flew at approximately the same height as their nearest leading neighbor (−0.01 ± 0.02 m, mean ± s.d. for the median trailing height across all 18 flocks).

Both nearest-neighbor lateral distance and front-back distance differed among flocks and species (*Figure 3a*). Species wingspan strongly predicted modal lateral neighbor position (linear regression, slope = 0.85, $R^2$ = 0.93, F = 228.29, p<0.0001). Wingspan also predicted front-back distance (slope = 0.70, $R^2$ = 0.86, F = 99.57, p<0.0001), although less strongly than lateral distance. After scaling alignment positions to wingspan (i.e., dividing neighbor distances by species wingspan), a distinctive pattern emerges (*Figure 3b*). Specifically, the flocks adopted a modal lateral distance of approximately one wingspan (mean 1.04, range 0.88–1.24 wingspans). This non-dimensionalized lateral distance had a weak inverse relationship to species wingspan (linear regression, slope = −0.37, $R^2$ = 0.37, F = 9.38, p=0.007) and was not related to flock density (i.e. nearest neighbor distance, non-dimensionalized by wingspan; linear regression, $R^2$ = 0.07, F = 1.15; p=0.30). Non-dimensionalized trailing distance was inversely proportional to species wingspan (linear regression, slope = −0.40, $R^2$ = 0.33, F = 7.93, p=0.012) and increased with non-dimensional flock density (linear regression, slope = 0.13, $R^2$ = 0.58, F = 22.39; p=0.0002). In

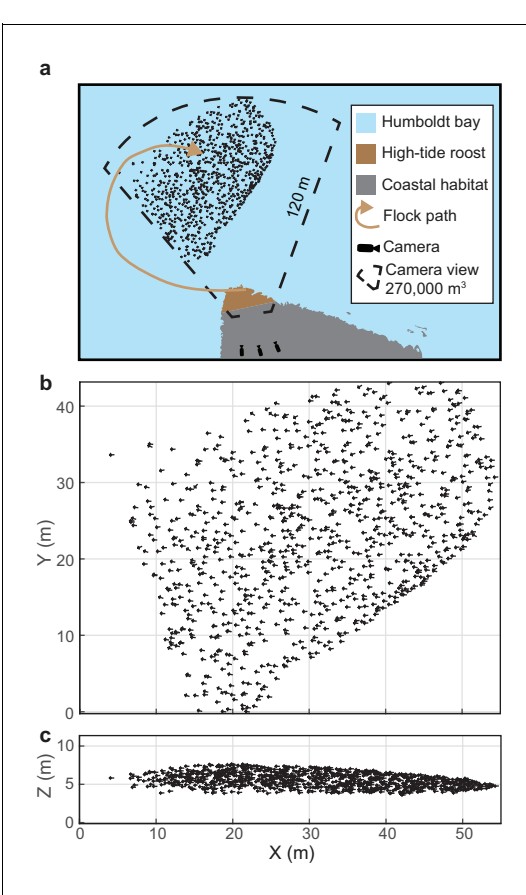

**Figure 1.** Shorebird flock recording. (**a**) Multi-camera videography was used to reconstruct 3D trajectories of shorebirds flying near high-tide roosts in Humboldt Bay, California. (**b**) Overhead and (**c**) profile views of an example flock. Symbol sizes reflect actual scales for birds with outstretched wings. Flock position data are available in *Figure 1—source data 1*.

DOI: https://doi.org/10.7554/eLife.45071.003

The following source data is available for figure 1:

**Source data 1.** Three-dimensional trajectory data for godwit flock 0420–1.

DOI: https://doi.org/10.7554/eLife.45071.004

**Table 1.** Flock parameters.

| Flock | Species | N birds | N frames | †,‡Nearest neighbor distance (m). | §Nearest neighbor power | †Ground speed (m·s⁻¹) | †Airspeed (m·s⁻¹) | Wind speed (m·s⁻¹) | ¶Wind direction (deg.) | †Z-speed (m·s⁻¹) | †Turnrate (° s⁻¹) |
|---|---|---|---|---|---|---|---|---|---|---|---|
| 0417–2 | Godwit | 286 | 147 | 1.30 (0.79–2.15) *1.67* | 0.361 | 5.23 (3.59–8.23) | 9.36 (8.04–10.54) | 5.73 | 4.3 | 1.11 (0.16–1.91) | 40.5 (18.4–80.9) |
| 0417–2 | Dowitcher | 309 | 147 | 1.16 (0.72–1.89) *2.23* | 0.385 | 5.15 (3.39–8.42) | 9.25 (7.94–10.47) | 5.73 | 4.63 | 1.26 (0.30–2.00) | 43.0 (19.4–81.9) |
| 0417–3 | Godwit | 474 | 278 | 1.81 (1.01–2.94) *2.32* | 0.428 | 3.32 (1.71–6.55) | 7.87 (6.42–9.39) | 5.73 | 5.3 | −0.18 (−1.27 – 0.76) | 26.8 (8.0–80.6) |
| 0417–4 | Godwit | 803 | 397 | 1.71 (1.02–2.58) *2.19* | 0.391 | 6.30 (3.04–9.44) | 8.82 (7.68–9.93) | 5.73 | 47.2 | 0.22 (−0.73 – 1.03) | 14.3 (4.66–40.8) |
| 0417–4 | Dowitcher | 74 | 397 | 1.57 (0.94–2.37) *3.01* | 0.382 | 6.51 (3.00–9.56) | 8.65 (7.49–9.65) | 5.73 | 48.1 | 0.39 (−0.53 – 1.21) | 19.8 (6.6–49.2) |
| 0420–1 | Godwit | 639 | 177 | 1.19 (0.59–1.94) *1.52* | 0.408 | 7.05 (5.12–9.91) | 10.54 (7.91–12.98) | 4.06 | 9.2 | −0.26 (−1.83 – 1.03) | 22.6 (7.39–75.9) |
| 0420–2 | Godwit | 309 | 147 | 1.54 (0.83–2.38) *1.97* | 0.43 | 8.95 (7.09–11.59) | 10.50 (8.76–12.11) | 4.06 | 59.7 | −0.06 (−0.95 – 0.68) | 11.9 (3.77–37.7) |
| 0427–2 | Dowitcher | 354 | 397 | 1.07 (0.61–2.00) *2.06* | 0.424 | 6.22 (3.06–9.76) | 9.03 (5.53–12.77) | 3.50 | 23.5 | 0.56 (−0.89 – 1.79) | 20.2 (6.68–60.1) |
| 0427–3 | Dowitcher | 391 | 217 | 1.15 (0.61–2.06) *2.21* | 0.463 | 10.9 (8.64–13.3) | 10.98 (7.99–13.98) | 3.50 | 73.3 | 0.32 (−2.10 – 2.15 | 24.8 (8.30–70.9) |
| 0427–5 | Dowitcher | 511 | 170 | 1.23 (0.70–1.98) *2.37* | 0.421 | 5.31 (4.23–6.49) | 7.14 (4.85–9.18) | 4.60 | 26.9 | 0.73 (−0.24 – 1.52) | 20.8 (7.62–57.4) |
| 1230–1 | Dunlin | 351 | 198 | 1.08 (0.58–1.78) *3.18* | 0.465 | 6.98 (6.03–7.85) | 7.65 (6.67–8.34) | 1.20 | 44.1 | −0.23 (−0.52 – 0.09) | 27.3 (11.9–46.6) |
| 1230–2 | Dunlin | 592 | 75 | 0.80 (0.47–1.23) *2.35* | 0.387 | 6.61 (6.00–7.30) | 7.60 (5.84–8.48) | 1.10 | 22.6 | −0.12 (−0.56 – 0.32) | 11.4 (3.7–26.8) |
| 1230–3 | Dunlin | 477 | 125 | 0.86 (0.49 1.37 *2.53* | 0.392 | 6.71 (5.85–7.73) | 7.54 (5.52–8.36) | 1.08 | 35.5 | 0.11 (−0.49 – 0.84) | 19.6 (5.27–42.9) |
| 1230–4 | Dunlin | 189 | 73 | 0.89 (0.50–1.50) *2.62* | 0.502 | 8.28 (7.44–9.70) | 7.47 (5.27–8.53) | 1.08 | 36.5 | −0.03 (−0.57 – 0.45) | 24.6 (7.5–52.9) |
| 0101–1 | Dunlin | 1039 | 228 | 1.03 (0.59–1.64) *3.03* | 0.41 | 8.39 (7.41–9.98) | 7.46 (5.74–8.64) | 1.63 | 118.4 | −0.23 (−0.73 – 0.27) | 17.7 (4.9–42.1) |
| 0101–3 | Dunlin | 961 | 188 | 0.92 (0.52–1.50) *2.71* | 0.416 | 8.61 (7.70–9.42) | 7.74 (6.25–8.76) | 1.63 | 117.6 | −0.02 (−0.33 – 0.36) | 13.0 (3.6–32.2) |
| 0101–4 | Dunlin | 269 | 340 | 1.00 (0.35–2.12) *2.94* | 0.45 | 6.56 (4.96–8.12) | 7.63 (5.12–8.72) | 1.63 | 39.3 | −0.14 (−0.70 – 0.30) | 18.3 (4.4–46.9) |
| 1220–1 | Avocet | 323 | 90 | 1.09 (0.70–1.69) *1.51* | 0.429 | 6.02 (4.55–7.57) | 8.18 (6.21–9.31) | 2.39 | 1.4 | 0.33 (−0.98 – 0.90) | 25.2 (10.4–48.8) |
| 1220–2 | Avocet | 321 | 245 | 1.19 (0.72–1.90) *1.65* | 0.432 | 6.96 (5.26–9.22) | 8.00 (6.88–8.96) | 2.39 | 8.2 | 0.10 (−1.49 – 0.74) | 30.2 (12.7–54.5) |

*Table 1 continued on next page*

*Table 1 continued*

| Flock | Species | N birds | N frames | †,‡Nearest neighbor distance (m). | §Nearest neighbor power | †Ground speed (m·s⁻¹) | †Airspeed (m·s⁻¹) | Wind speed (m·s⁻¹) | ¶Wind direction (deg.) | †Z-speed (m·s⁻¹) | †Turnrate (° s⁻¹) |
|---|---|---|---|---|---|---|---|---|---|---|---|
| 1220–3 | Avocet | 281 | 280 | 1.30 (0.78–2.10) *1.81* | 0.472 | 7.50 (5.32–8.93) | 7.93 (6.28–8.88) | 2.39 | 22.3 | 0.33 (−0.62–0.83 | 23.9 (6.18–49.1) |

†Values are medians and (in brackets) $10^{th}$-$90^{th}$ percentiles of values extracted at one-wingbeat intervals from all individuals of each flock.

‡Values in italics are in wingspan units instead of metric units.

§Exponent of power law fit to distance of 10 nearest neighbors.

¶Wind direction is relative to the overall flight direction where 0° is a pure headwind and 180° a pure tailwind.

Note that data are presented separately in consecutive rows for each species in mixed-species flocks (0417–2 and 0417–4). Data used for generating this table are available in *Table 1—source data 1*.

DOI: https://doi.org/10.7554/eLife.45071.005

The following source data is available for Table 1:

Source data 1. Flock parameter data.

DOI: https://doi.org/10.7554/eLife.45071.006

summary, across all four species, shorebirds adhere to a non-dimensional spacing rule of aligning to neighbors with a lateral offset of approximately one wingspan while allowing trailing distance to vary with flock density.

Data from mixed-species flocks of godwits and dowitchers further support the non-dimensional nature of the lateral spacing rule within individual flocks. Both dowitchers and godwits adjusted their lateral spacing depending on the species of their neighbor (*Figure 4*). Godwits following conspecifics had a modal lateral spacing of 0.76 m, or 0.97 godwit wingspans. When following the smaller dowitchers, godwits reduced the modal lateral distance to 0.60 m or 0.92 wingspans when calculated using the average wingspan of dowitchers and godwits (Mann-Whitney $U$ = 525,684; $n_1$ = 1034; $n_2$ = 81; p=0.0004). Dowitchers following conspecifics flew with a modal lateral distance of 0.51 m, or 0.98 dowitcher wingspans. When following the larger godwits, dowitchers increased the modal lateral distance to 0.58 m or 0.89 average wingspans (Mann-Whitney U test; U = 66,341; $n_1$ = 743; $n_2$ = 149; p<0.0003).

## Comparison of simple- and compound-V formations

While recording the larger cluster flocks, we also recorded four godwit simple-V formations of between 16 and 44 individuals, which were recorded for between 42 and 211 frames (*Figure 5*). Here we compare the positioning of godwits in simple and compound-V formations. In both cases, nearest neighbors were most commonly in the same horizontal plane (mean of 61% in godwit cluster flocks, 97.9% in godwit simple-V formations), defined as extending one wingspan above and below the focal bird, with the follower positioned over a narrow lateral range and wider range of trailing distances (*Figures 3b* and *5b*). The modal lateral position in the simple-V formations was slightly less (mean of 0.8 wingspans) than that in the compound-V formations, where the mean modal lateral position among godwit flocks was 0.96 wingspans (Generalized Linear Model with terms for flock and simple versus compound-V formation; p<0.0001; *Figure 5*, *Figure 5—source data 1*). The modal trailing distance in simple-V formations was 0.50 wingspans; in compound-V formations of godwits, the mean modal trailing distance was 0.86 wingspans.

## Extended flock structure

We next examined how individual neighbor alignment rules relate to flock structure. We measured the angular distribution of neighbors at distances of two, four, six, and eight wingspans and at the maximum distance at which half of the flock remains in the flock's core (range 5.8–24.3 wingspans). This last measure was used as a proxy for whole flock structure while avoiding edge effects (see 'Materials and methods'). For this analysis, we included all neighbors flying within a ±15 degree elevation slice relative to each focal bird. This was used instead of the ±1 wingspan slice used in other analyses (e.g., *Figure 2*, *Figure 3*) because this metric corresponds to a decreasing proportion of the volume at further distances. At a distance of two wingspans, flocks were consistently

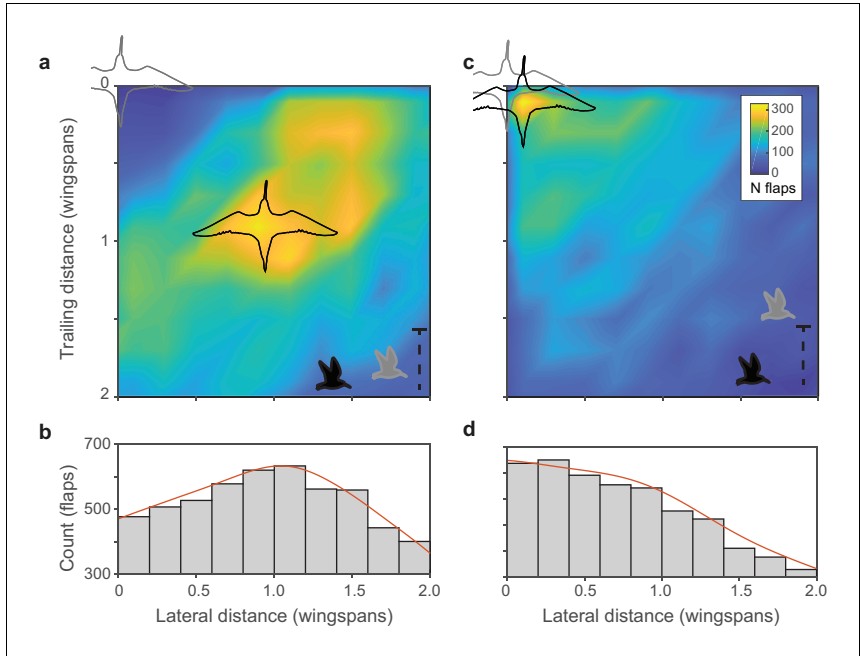

**Figure 2.** Within-flock positioning. (**a, b**) Histograms of nearest-neighbor alignment for birds flying within ±1 wingspan of elevation (godwit flock 0420–1) show a distinctive peak at a trailing distance and lateral distance of approximately one wingspan; focal birds are shown in light gray and nearest neighbors in black. Inset bird silhouettes show profile views of the birds' relative flight elevations. (**c, d**) Histograms of nearest-neighbor alignment for birds flying outside ±1 wingspan of elevation for the same flock show a largely random distribution with a modal location of nearly straight above or below the focal bird. Data used for generating this figure are available in *Figure 2—source data 1*.

DOI: https://doi.org/10.7554/eLife.45071.007

The following source data is available for figure 2:

**Source data 1.** Nearest-neighbor positioning data for flock 0420–1.
DOI: https://doi.org/10.7554/eLife.45071.008

---

asymmetrical, with trailing birds more frequently flying to the left of their leading neighbors in 12 of 18 flocks and to the right of their nearest leading neighbors in the remaining six flocks. This asymmetry persisted at all distances within the flock (*Figure 6a*), including the overall flock shape (*Figure 6b*). The direction of asymmetry was independent of relative camera viewing direction and flock turning direction but was positively correlated with relative wind direction (see statistical results in *Table 2*).

## Flock biomechanics

We quantified several biomechanically relevant parameters from individual birds in flocks, including ground speed, estimated air speed, ascent or descent speed, wingbeat frequency and flapping phase. We created linear mixed effects (LME) statistical models to predict wingbeat frequency and airspeed from local and global flock positions and other flight parameters (*Table 3*). While speeds were measured for all individuals, flapping frequency and phase were only available from six cluster flocks in which birds were sufficiently close to the cameras to allow wingbeat measurements and for the simple-V formations. We examine only data for which wingbeat and estimated air speed data were available (N = 3306 individuals). We were also unable to measure flapping parameters from Dunlin, the smallest species recorded here.

We observed several individual and flock effects on flight speed and wingbeat frequency (see *Table 3* for full statistical results). As expected, different species flew with different characteristic flapping frequencies (LME, p<0.00001 for all species) and speeds (LME, p<0.00001 for dowitcher and avocet, p=0.00022 for godwit), and climbing flight was associated with an increase in flapping

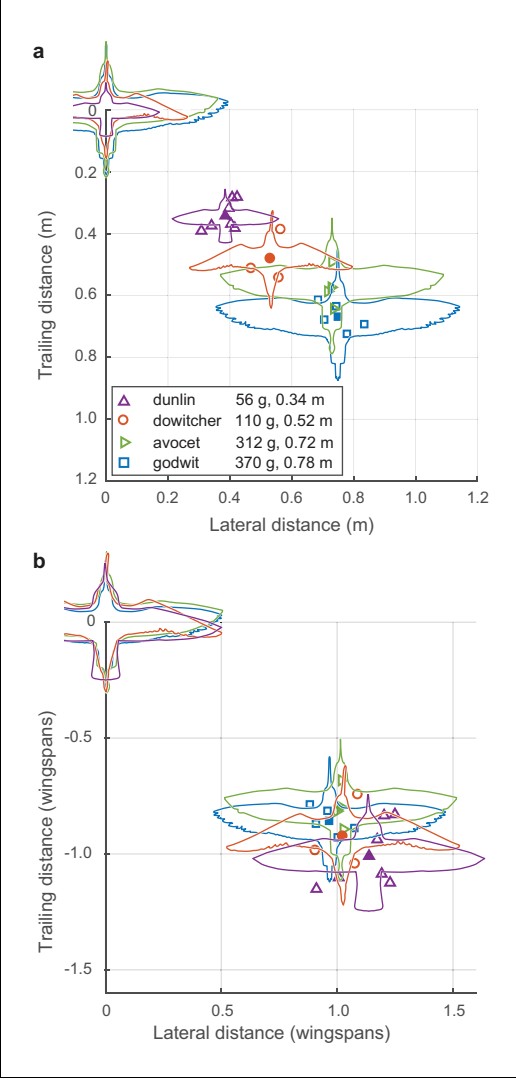

**Figure 3.** Modal positioning among flocks and species. (a) Summary of modal neighbor position for nearest neighbors within ± 1 wingspan in single-species flocks of all four species, depicted in absolute metric distance and (b) the same data plotted in distances relative to the wingspan of each species. Open symbols indicate modal neighbor positions for individual flocks. Closed symbols and silhouettes show the average position for each species. Data used for generating this figure are available in *Figure 3—source data 1*.

DOI: https://doi.org/10.7554/eLife.45071.009

The following source data and figure supplements are available for figure 3:

**Source data 1.** Modal neighbor positioning data for all shorebird flocks and for a flock of chimney swifts.

DOI: https://doi.org/10.7554/eLife.45071.012

**Figure supplement 1.** Distribution of nearest neighbors in the horizontal slice (±1 wingspan elevation) from a chimney swift roosting flock (*Evangelista et al., 2017*).

DOI: https://doi.org/10.7554/eLife.45071.010

frequency (LME, p<0.00001). Birds flying near the front of the flock along the direction of travel (birds were given a continuous index with 0 being the frontmost and 1 the rearward-most position) flew faster (LME, p<0.00001) and with a lower flapping frequency than those near the rear (LME, p<0.00006). Birds flying near the edge of the flock also flew faster than those in the middle (LME, p<0.00001). Higher flapping frequencies were correlated with slower flight (LME, p<0.00001). Birds flying within a plausible range of locations for aerodynamic interaction (0.7–1.5 wingspans lateral distance and within two wingspans overall distance of their leading neighbor, coded as 'Aerodynamic neighbor' in *Table 3*) flew faster than expected after controling for the other effects described above (LME, p<0.00001, *Figure 7*). However, positioning in this aerodynamic interaction region had no effect on flapping frequency, as it was not in our best model of wingbeat frequency based on Bayesian information criteria (BIC; *Table 3*). Adding the aerodynamic neighbor to the best model makes the term non-significant (LME, p=0.30) and increases the BIC of the model by 6.8.

We examined the compound-V-flock data for evidence of flapping synchronization by examining the temporal and spatial phase offset between pairs of nearest neighbors for which synchronous wingbeat frequency data were available for at least 20 frames (see 'Materials and methods'). We found no evidence for temporal (Rayleigh test; N = 117; Z = 1.98; p=0.14) or spatial wingbeat synchronization (Rayleigh test; N = 117; Z = 1.28; p=0.27) in the compound-V-formation shorebird flocks. We performed the same tests on the simple-V formation of godwits and again found no support for phasing relationships (Rayleigh test; temporal phasing N = 39; Z = 0.09; p=0.90; spatial phasing; N = 39; Z = 0.46, p=0.63).

## Discussion

Here, we report on the first cross-species quantitative analysis of bird flocking behavior. On the basis of previous studies, we predicted that larger species would adopt more structured flocks and would exhibit more frequent aerodynamic positioning. Neither of these hypotheses were supported by our data. Instead, we document a flock structure that we term the compound-V formation, in which birds in cluster flocks align to nearest neighbors within their same elevation slice (±1 wingspan) with a one-wingspan lateral offset while allowing front-back

*Figure 3 continued*

**Figure supplement 2.** Distribution of nearest neighbors in the horizontal slice (±1 wingspan elevation) from all shorebird flock data described here, regardless of species.
DOI: https://doi.org/10.7554/eLife.45071.011

distance to fluctuate with flock density (*Figures 2, 3* and *4*). This flock type is similar to the shorebird 'cluster' and 'bunch' formations described by *Piersma et al. (1990)* and the 'front cluster' of *Heppner (1974)*. Here, this structure was observed in single- and mixed-species flocks of four shorebird species covering a seven-fold range in body mass. The simple alignment rule produces a flock structure that can be observed at all spatial scales within the flock, including overall flock shape (*Figure 6*). This is in contrast to flocks of other species, such as starlings, in which structure is only observed within each neighbor's six nearest neighbors, equivalent to 1.2–2.7 wingspans (*Ballerini et al., 2008a*). Our data also show that shorebird global flock alignment is responsive to estimated local wind conditions (*Table 2*), and future work exploring this interaction may allow identification of the mechanism that governs the overall alignment. Wind conditions did not have discernible effects on local alignment, possibly because of the uncertainty in the measurement of the wind vector itself.

Mixed-species assemblages typically represent around 10% of migratory shorebird flocks (*Piersma et al., 1990*), possibly because species that have different preferred flight speeds would have to compromise their flight speed in order to remain together as a group. Our data include ~10% mixed-species flocks (2 of 18) and support the hypothesis that differences in preferred flight speed influence whether different species flock together. We documented two mixed-species flocks of godwit and dowitcher; these flocks had an airspeed of 9.02 ± 0.34 m s$^{-1}$ (mean ± s.d., n = 2). The single-species godwit and dowitcher flocks had airspeeds of 9.64 ± 1.53 (n = 3) and 8.99 ± 1.92 m s$^{-1}$ (n = 3), respectively. Dunlin, which were present at the same time as godwits and dowitchers but did not fly in mixed flocks with these species, had an airspeed of 7.58 ± 0.11 m s$^{-1}$ (n = 7). Similarly, Avocets not observed to mix with other species at our field site, and they flew with airspeeds of 8.04 ± 0.13 m s$^{-1}$ (n = 3). Thus, the similarity in preferred flight speeds among godwits and dowitchers might be important for these species to form mixed-species flocks. Dunlin and avocets also flew at similar airspeeds, but were not observed in mixed-species flocks, perhaps because of their large difference in body size, wingbeat frequency, and/or maneuverability.

The flock data presented here also include other interesting results that lack clear explanations. Flight speed varied with position from front to rear and from center to margin (*Table 2*), implying that the flocks were not necessarily in equilibrium. This might cause larger flocks to separate into several smaller flocks over time, consistent with the observation that the arrival group size of migratory species is typically smaller than the departure group size (*Piersma et al., 1990*).

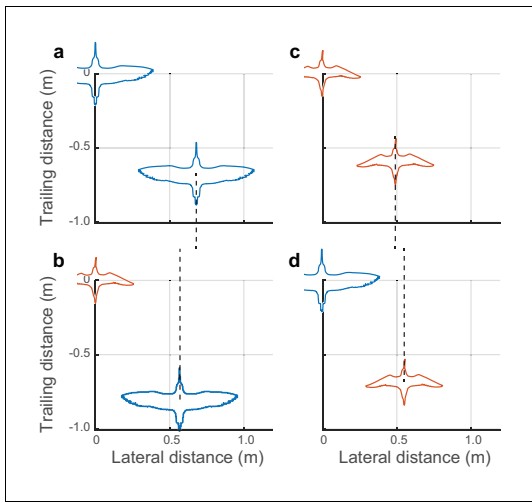

**Figure 4.** Positioning in mixed-species flocks. Data from mixed species flocks show that birds adjust their lateral spacing depending on the species (and size) of their nearest leading neighbor. (**a**) Godwits following conspecifics adopt a larger lateral distance than (**b**) godwits following the smaller dowitchers. (**c**) Dowitchers following conspecifics use a shorter lateral distance than (**d**) dowitchers following the larger godwits. These results support the hypothesis that shorebirds adopt a lateral spacing rule that is dependent on the size of their leading neighbor. Dashed lines are provided to facilitate comparison of modal lateral positions between (**a**) and (**b**) and between (**c**) and (**d**). Data used to generate this figure are available in *Figure 4—source data 1*.
DOI: https://doi.org/10.7554/eLife.45071.013
The following source data is available for figure 4:

**Source data 1.** Neighbor position data for mixed-species flocks.
DOI: https://doi.org/10.7554/eLife.45071.014

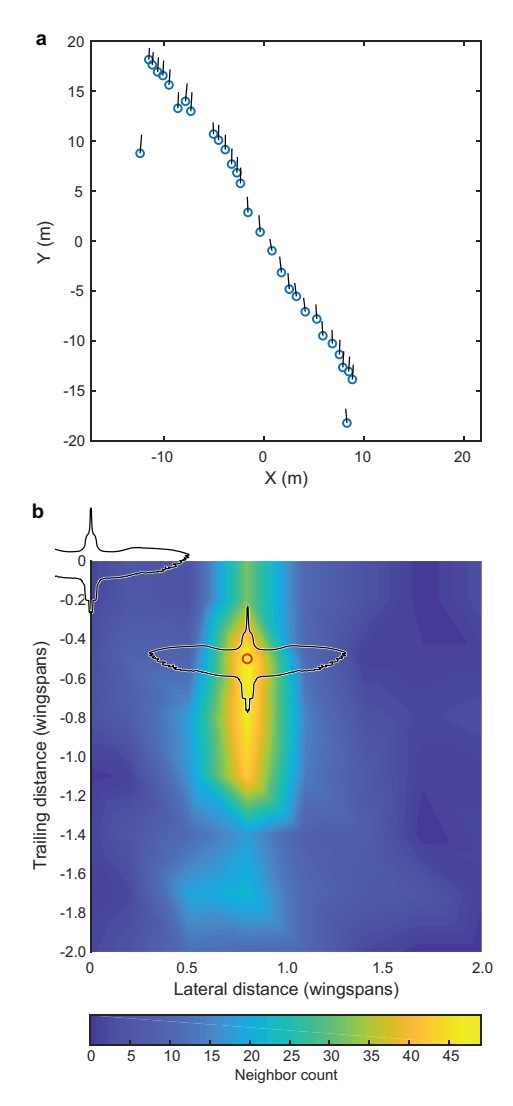

**Figure 5.** Godwit simple-V formation. Incidental to our cluster flock recordings, we also recorded several instances of godwits flying in a simple-V, echelon or line formation, and the largest of these examples is shown here. (a) Overhead view of the flock; average flight direction is along the positive Y axis; blue circles show bird positions and black lines are 2D velocity vectors. All birds are within a ± 1 wingspan horizontal slice. (b) The relative location of nearest neighbors; the modal location (red circle) was at a displacement of 0.8 wingspans lateral and 0.5 wingspans trailing distance. Trailing position was more varied than lateral position. Wind speed was low (<2 m s$^{-1}$) according to weather station data and the wind speed estimated from the ground speed and flight direction of the birds.

The data used to generate this figure are available in *Figure 3—source data 1*.

DOI: https://doi.org/10.7554/eLife.45071.015

The following source data is available for figure 5:

**Source data 1.** Godwit simple-V-formation position data.

DOI: https://doi.org/10.7554/eLife.45071.016

Because birds in simple and compound-V formations adopt similar neighbor alignment rules, functional hypotheses for simple-V formations might also apply to compound-V formations. These include collision avoidance and information transfer (*Dill et al., 1997*). Collision avoidance is a plausible hypothesis to explain the formation of simple-V formations because they theoretically permit birds to keep all neighbors out of their direct path of travel. This is not the case for compound-V formations, where many birds are flying in front of and behind one another (*Figure 1b*; *Figure 6b*). The problem of collision avoidance is exacerbated in compound-V formation because birds tend to fly in the same horizontal plane. A better strategy for collision avoidance is to fly in a three-dimensional shape, such as that adopted by flocks of chimney swifts (*Evangelista et al., 2017*). In these flocks, the most common neighbor position is further lateral than in the shorebird flocks and with a shorter trailing distance, more completely moving those individuals out of the path of other flock members (*Figure 3—figure supplement 1*). Finally, even in the simple-V formation recorded here (*Figure 5*), birds flew with approximately 20% of wingspan overlap and so did not have an entirely clear forward path. Thus, collision avoidance appears to be an unlikely explanation for the structuring of both the compound-V and simple-V formations recorded here.

Simple and compound-V formations might also be structured to maximize the observability of neighbors, facilitating information transfer by helping birds to detect and respond to changes in neighbor speed or direction, and improving flock cohesiveness by allowing information to propagate through the flock more quickly. *Dill et al. (1997)* proposed that birds in V formation should maximize the measurement of neighbor movements by aligning at a 35.3 degree angle (relative to the direction of travel), or alternatively should maximize the measurement of neighbor speed by aligning at a 63.4 degree angle. The shorebird flocks examined here had modal neighbor-position alignment angles ranging from 33.7 to 51.8 with an average of 41.2 degrees. Neither this mean angle nor the nearly 20-degree range in alignment angle is consistent with Dill's hypotheses or others calling for a single optimal alignment angle. Our finding that lateral spacing is uncorrelated with flock density, whereas trailing spacing increases with decreasing density, shows that the shorebird flocks are more organized in lateral distance than in trailing distance or alignment angle. Thus, hypotheses

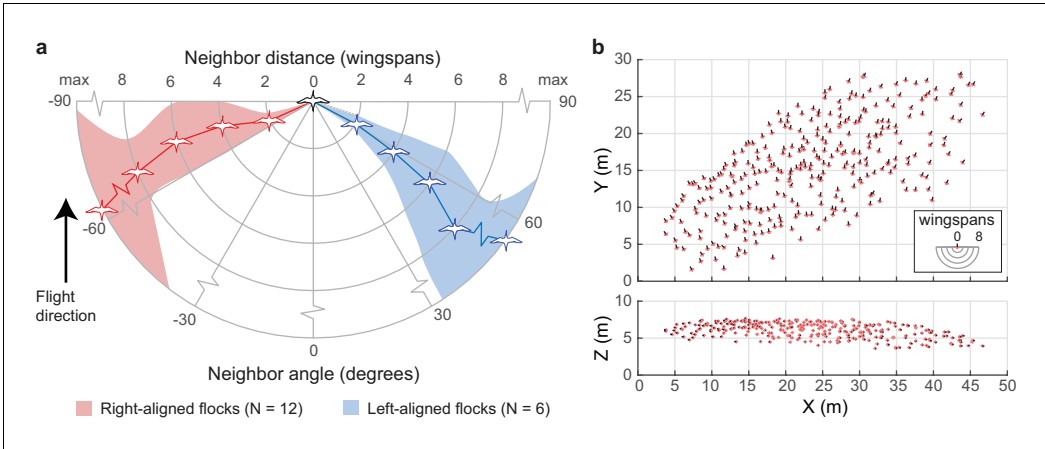

**Figure 6.** Extended flock structure. (**a**) Polar plot showing mean neighbor angle for right-aligned and left-aligned flocks over a range of distances. Shaded regions show 95% confidence intervals. (**b**) Overhead and profile views of an example right-aligned flock (avocet flock 1220–2). Note the many echelon formations aligned from back left to front right and the overall shape of the flock. The inset shows scale in wingspans. The data used to generate this figure are available in *Figure 6—source data 1*.

DOI: https://doi.org/10.7554/eLife.45071.017

The following source data is available for figure 6:

**Source data 1.** Extended flock structure data.

DOI: https://doi.org/10.7554/eLife.45071.018

calling for organization based on alignment angle, whether to maximize information transfer or to keep lead birds in the visual fovea of trailing neighbors in a V formation (**Badgerow and Hainsworth, 1981**), are not well supported by our results.

Theoretical (**Badgerow and Hainsworth, 1981**; **Hummel, 1983**; **Lissaman and Shollenberger, 1970**; **Maeng et al., 2013**) and empirical research (**Portugal et al., 2014**; **Weimerskirch et al., 2001**) has provided support for the hypothesis that birds flying in simple-V formations gain aerodynamic and energetic benefits, and we propose that such benefits might also explain why birds adopt the compound-V formation. In both cases, birds fly with a lateral offset of approximately one wingspan while allowing trailing distance to vary (**Figures 3** and **5**), facilitating aerodynamic interaction. When compared to simple-V formations, compound-V formations allow greater flock densities, which should allow more rapid information transfer (**Attanasi et al., 2014**), larger flock sizes, and improved predator defense (**Powell, 1974**). Analyses of the airspeeds and wingbeat frequencies of flocking shorebirds provide some support for the aerodynamic alignment hypothesis. Birds flying in positions where beneficial aerodynamic interactions are predicted to occur flew faster than expected after controling for other factors (aerodynamic neighbor term in **Table 3**, linear mixed effects model

**Table 2.** Flock orientation.

| Variable | Test | N | $R^2$ | t/F | P |
| --- | --- | --- | --- | --- | --- |
| Wind direction | Circular correlation | 18 | 0.29 | 2.29 | 0.02 |
| Turn direction | Linear regression | 18 | 0.00 | 0.04 | 0.83 |
| Camera direction | Circular correlation | 18 | 0.02 | 0.61 | 0.54 |

Tests of the relationship between flock left-right orientation (**Figure 6**) and environmental factors. The data usedto generate this table are available in **Table 2—source data 1**.

DOI: https://doi.org/10.7554/eLife.45071.019

The following source data is available for Table 2:

Source data 1. Flock orientation data.

DOI: https://doi.org/10.7554/eLife.45071.020

**Table 3.** Flock biomechanics.

| Wingbeat frequency predictors | Estimate | S.E. | T | d.f. | P |
|---|---|---|---|---|---|
| Intercept (dowitchers) | 8.82 | 0.132 | 66.5 | 2817 | <0.00001 |
| Godwit | −2.19 | 0.035 | −63.5 | 2817 | <0.00001 |
| Avocet | −1.91 | 0.040 | −47.5 | 2817 | <0.00001 |
| Airspeed (m s$^{-1}$) | −0.05 | 0.013 | −4.3 | 2817 | 0.00002 |
| Flock position | 0.13 | 0.032 | 4.0 | 2817 | 0.00006 |
| Nearest neighbor distance (wingspans) | −0.03 | 0.010 | −3.2 | 2817 | 0.00153 |
| Nearest neighbor species | −0.45 | 0.038 | 11.9 | 2817 | <0.00001 |
| Z-speed (m s$^{-1}$) | 0.29 | 0.019 | 15.1 | 2817 | <0.00001 |
| **Airspeed predictors** | **Estimate** | **S.E.** | **T** | **d.f.** | **P** |
| Intercept (dowitchers) | 10.69 | 0.225 | 47.5 | 2832 | <0.00001 |
| Godwit | −0.25 | 0.067 | −3.7 | 2832 | 0.00022 |
| Avocet | −1.51 | 0.067 | −22.6 | 2832 | <0.00001 |
| n.n. distance (wingspans) | −0.08 | 0.015 | −5.2 | 2832 | <0.00001 |
| Edge distance (wingspans) | −0.03 | 0.003 | −10.3 | 2832 | <0.00001 |
| Wingbeat frequency (H z) | −0.12 | 0.025 | −5.0 | 2832 | <0.00001 |
| Flock position | −0.24 | 0.045 | −5.4 | 2832 | <0.00001 |
| Aerodynamic neighbor | 0.23 | 0.040 | 5.7 | 2832 | <0.00001 |

Results from linear mixed-effects models relating wingbeat frequency and airspeed to other measured variables. Nearest neighbor only defined when this bird is leading the focal bird. Godwit and avocet are dummy variables coding species differences relative to dowitchers. Nearest neighbor species is coded −1 for a smaller neighbor, 0 for same the species, and 1 for a larger neighbor. Flock position is continuously scaled from 0.0 (front) to 1.0 (back). Aerodynamic neighbor was coded 1 for birds flying within 0.7–1.5 wingspans lateral distance and within two wingspans distance from their nearest leading neighbor, 0 otherwise. Models were selected using Bayesian information criteria. The data used to generate this table are available in *Table 3—source data 1*. d.f., degrees of freedom.

DOI: https://doi.org/10.7554/eLife.45071.021

The following source data is available for Table 3:

**Source data 1.** Flock biomechanical data.

DOI: https://doi.org/10.7554/eLife.45071.022

for airspeed, p<0.0001). Over the entire dataset, 29.7% of nearest neighbor positions were in the 'aerodynamic neighbor' location (*Figure 3—figure supplement 2*), compared with only 3.4% of nearest neighbor in flocking chimney swifts (*Figure 3—figure supplement 1*). This faster flight should produce a reduced cost of transport, assuming there are no unmeasured compensating factors such as a simultaneous increase in stroke amplitude. Nevertheless, this speculative interpretation of the compound-V formation raises many new questions, such as how birds in the flock can maintain different speeds without separating and why an aerodynamic benefit would manifest as an increase in speed instead of, for example, a reduction in flapping frequency and airspeed as suggested by theoretical models (*Hummel, 1983*). Furthermore, despite the similarities in modal position among compound-V and simple-V flocks (*Figures 3* and *5*), it is not clear whether a single set of adjustment rules or responses to changes in neighbor position can produce both flock types. These questions, and a definitive explanation for why birds adopt a compound-V formation, cannot be answered with the current dataset. Progress in these areas will depend on new theoretical modeling and data collection from on-bird loggers measuring physiological, flock positioning and biomechanical data from a variety of species over a range of behavioral contexts. Further videographic flock surveys may also improve understanding of the variety of flock types, especially when collected with careful attention to behavioral context and with full measurement of local environmental conditions.

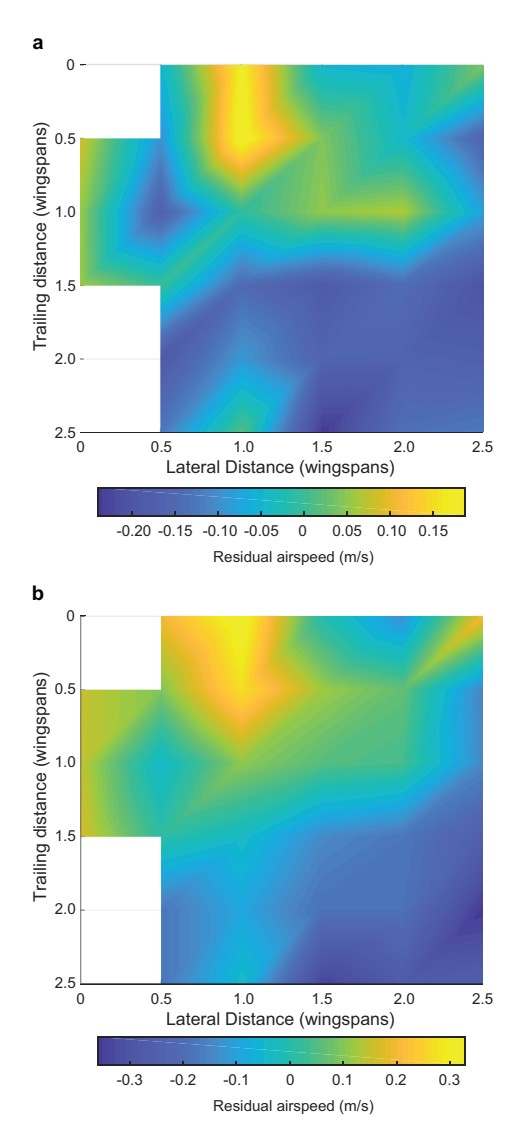

**Figure 7.** Effect of positioning for aerodynamic interaction. Here we show the effect of neighbor position on flight speed. (a) Flight speed residuals after accounting for species, flapping frequency, distance from flock edge, nearest neighbor distance in terms of wingspans and overall position along the length of the flock. (b) Flight speed residuals after accounting for just species and flapping frequency. White spaces in the heat map are bins with fewer than 20 samples, out of 2848 possible in (a) and 3306 possible in (b). Both analyses reveal a broadly similar pattern, where the positive effect of neighbor position on flight speed is strongest at a one wingspan lateral displacement and a trailing distance of 0 to 0.5 wingspans. This pattern cannot be generated by trailing birds passing leaders because the roles reverse after passing occurs, leaving no net speed difference. The data used to generate this figure are available in *Figure 7—source data 1*.
DOI: https://doi.org/10.7554/eLife.45071.023

*Figure 7 continued on next page*

## Materials and methods

### Field recording

We recorded multi-camera video of freely behaving, wild birds in Humboldt County, California between 17 April and 27 April 2017 and between 20 December 2017 and 1 January 2018. Recordings were made at the Arcata Marsh Wildlife Sanctuary (40°51′25.35"N, 124° 5′39.37"W) and above agricultural fields in the Arcata bottoms (40°53′51.98"N, 124° 6′55.85"W). No birds were captured or handled, and we made efforts to avoid influencing bird behavior. Video was captured at 29.97 frames per second and 1920 × 1080 pixel resolution using three Canon 6D cameras with 35 mm or 50 mm lenses. Cameras were set along a 10 m transect and staggered in elevation. We set cameras up to overlook locations where birds aggregated during high tide or when foraging in agricultural fields. Flocking events included birds moving with the tide, or flushing in response to predators (e.g., peregrine falcons) or for unknown reasons. Cameras recorded continuously for up to 3 hr per day. For analysis, we selected flocks that included at least 100 individuals and that had an orientation and size that allowed visual discrimination of individuals within the flock.

### Bird detection

We used the MATLAB R2017a (Natick, MA, USA) computer vision toolbox to generate code for detecting birds in video recordings. A foreground detector first separated moving objects from the stationary background. A gaussian filter was then applied to the image with a diameter matched to bird size under the recording conditions. Two-dimensional peak detection found local peaks in the smoothed image that were taken as potential bird positions.

Under some conditions, overlapping wings of adjacent birds prevented accurate detection of many individuals. To overcome this problem, we developed a frame-averaging algorithm that helped to obscure the wings and to emphasize the bodies. Here, optic flow determines the overall movement of the flock for each frame. Using the optic flow measurements and two-dimensional interpolation, the algorithm subtracts movement between frames. A rolling 5-frame window is then applied to the entire video. This procedure highlights pixels that are moving in the same direction as the flock, such as the birds' bodies, while filtering pixels that are moving in other directions, such as the wings.

*Figure 7 continued*

The following source data is available for figure 7:

**Source data 1.** Neighbor position and flight speed data.
DOI: https://doi.org/10.7554/eLife.45071.024

## Three-dimensional calibration

Camera calibration followed established methodology (*Hedrick, 2008*; *Jackson et al., 2016*; *Theriault et al., 2014*), with the exception that the distance between cameras, instead of an object placed in view of the cameras, was used to scale the scene. This approach allowed us to record in locations where it was infeasible to place calibration objects in front of the cameras (e.g., over water). The in-camera horizontal alignment feature was used to align cameras to the horizon. The pitch of the camera was measured with a digital inclinometer with 0.1-degree precision. This allowed alignment of the scene to gravity in post processing, with the vertical (Z axis) origin placed at the level of the cameras. This permitted direct measurement of the elevation of the birds relative to one another.

Background objects that were visible in the scene were used as calibration points. We developed a preliminary calibration using stationary objects such as trees, poles, and sitting birds. We then added flying birds, ensuring that points covered a wide range of distances and elevations relative to the cameras. Calibrations had low direct linear transformation (DLT) residuals (<0.5–1 pixel), indicating high-quality calibrations.

## Camera synchronization

Cameras were synchronized by broadcasting audio tones over Walkie Talkies (Motorola Talkabout MH230) to each camera. Audio tones were broadcast approximately once every five minutes during recording. A time offset was determined for each pair of cameras using cross-correlation of the audio tracks. This offset allowed camera synchronization within ±one half of a frame, or 16.6 ms.

In recordings where birds were relatively close to the camera (<50 m) and moving at relatively high pixel speeds, we used sub-frame interpolation to achieve increased synchronization accuracy of one tenth of a frame, or ±1.7 ms. To determine the subframe offset, we interpolated tracks of moving birds used as background points in the calibration at 0.1 frame intervals from −1 to +1 frame (−1.0,–0.9, etc). We then calculated the DLT residual for a calibration with each combination of sub-frame-interpolated points for the three cameras. The set of offsets generating the lowest DLT residuals was used for the final calibration and applied to birds tracked in the study.

## Three-dimensional assignment

To reconstruct the three-dimensional positions of birds in a flock, 2D detections of individuals must be correctly assigned between cameras. We modified established software for this task (*Evangelista et al., 2017*; *Wu et al., 2009*). Briefly, the software first finds all combinations of 2D points having DLT residuals of <3 pixels. The software iteratively generates 3D points, starting with points that have the lowest DLT residuals and only allowing a 2D detection to be reused a single time. This helps with the problem of occlusion while limiting the number of 'ghost' birds (bird positions created from incorrectly matching detections among cameras). This process is repeated twice. The first iteration allows the user to determine a bounding region in the 3D space in which the flock is contained. In the second iteration, three-dimensional positions outside this bounding region are filtered before they can be considered as potential 3D points.

## Track generation

After 3D points have been generated, they are linked between frames to generate individual flight tracks. Here, a Kalman filter predicts the position of each bird in the subsequent frame for the 2D information from each camera and for the reconstructed 3D positions. In the first frame, the Kalman filter is seeded using optic flow measurements. For each frame step, a cost matrix is created from weighted sums of the 2D and 3D errors between predicted track positions and each reconstructed 3D point. The Hungarian algorithm is used to find a global optimum that minimizes the error in track assignment. A track that is not given an assignment is continued with a gap of up to four frames (0.13 s), after which it ends and any re-detection of the bird in question will start a new track.

## Wingbeat frequency analysis

We measured wingbeat frequencies in a subset of recordings in which birds were both large enough and close enough to cameras to discern wingbeat oscillations. This excluded flocks of our smallest species, dunlin, and some flocks that were relatively distant from cameras. To measure wingbeat frequency, we used blob analysis to find a bounding box for each bird in each frame. We excluded blobs for which the bounding box included two or more birds as determined using the track-assignment algorithm described above. We averaged four components of the bounding box to measure wingbeat phase: height, inverse of the width, detrended X-coordinate of top-left corner, and inverse of detrended Y-coordinate of the top-left corner. This allowed quantification of wingbeat phase independent of bird orientation with respect to the cameras. Wingbeat phase was averaged across cameras and bandpass filtered before a 128-point Fast Fourier Transform (FFT) was applied to measure wingbeat frequency. The frame rate of the cameras (29.97 frames per second) and the FFT window determined a wingbeat frequency bin size of 0.12 wingbeats $s^{-1}$. Our method is similar to that used in a recent study of two corvid species (*Ling et al., 2018*).

## Species identification in mixed-species flocks

We recorded two mixed-species flocks of godwits and dowitchers. The size difference between species allowed species identification using the detected pixel area and distance of each bird (*Figure 8*). Here, blob analysis quantifies the pixel area for each bird in each tracked frame. Area data were excluded when two tracked birds were within a single blob bounding box. A low-pass filter was applied to the sequence of pixel area data across frames for each tracked bird to remove wingbeat effects. An object's pixel area scales with the inverse of the square root of distance. Therefore, for each frame, the square root of the filtered pixel area was multiplied by bird's distance to provide a distance-scaled area. This value was averaged across frames and across cameras for each bird track. In mixed-species flocks, a histogram of the scaled area had two distinct peaks with only a small amount of overlap (*Figure 8a*). Fitting two normal distributions to these data revealed an expected error rate in species identification of 3.3%. The scaled area where the two normal distributions intersect was used as the threshold for species identification.

## Neighbor alignment metrics

We quantified the relative position of each bird and its nearest neighbor in the flock (*Figure 2*). This was done separately for neighbors within ±1 wingspan in flight elevation—the potential positions at which aerodynamic interactions and collisions are plausible—and for neighbors beyond ±1 wingspan. For each flock, we calculated the modal lateral distance and modal front-back distance by taking the peak of a probability density function generated with a kernel density estimator and a smoothing bandwidth of 0.25 wingspans. We used modal values because distance calculations are truncated at zero, producing skewed distributions.

In a subsequent analysis (*Figure 6*), we quantified the angular distribution of neighbors at

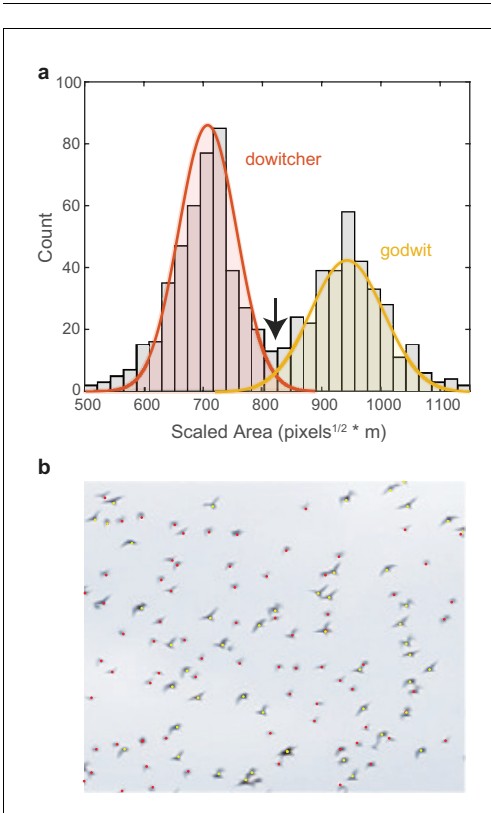

**Figure 8.** Species identification in mixed flocks. (**a**) Histogram of scaled pixel area of birds within a mixed-species flock. The two peaks are modeled as normal distributions. The area value where the two distributions intersect (indicated by the arrow) is used as the threshold for species identification. (**b**) Example section of a mixed-species flock with species identifications labeled by color.
DOI: https://doi.org/10.7554/eLife.45071.025

distances of two, four, six, and eight wingspans, and at a maximum radius depending on the size of the flock. Two-wingspan bins centered at the reference distance were used for selecting data points (e.g. birds within 1–3 wingspans were included in the two-wingspan bin). Our aim was to examine the extent of internal structure within the flock. Because edge effects could create the appearance of internal structure, we excluded birds whose edge distance was less than the wingspan of the bin being analyzed. For example, for the two-wingspan analysis, all birds within three wingspans of the horizontal edge of the flock were excluded. The maximum radius was taken as the median horizontal edge distance of all birds in the flock (*Figure 9*). This ensured that our analysis always included at least half of the flock.

## Wingbeat phase analysis

We conducted an analysis to test for temporal and spatial wingbeat phase synchronization, following previously established methods (*Portugal et al., 2014*). We selected pairs of nearest neighbors in flocks where simultaneous wingbeat frequency data were available for both individuals for at least 20 frames (0.66 s). Cross correlation was used to determine the temporal phase offset between the birds. This value was divided by $2\pi d$, where $d$ is wingbeat duration, to attain a value between 0 and $2\pi$. The spatial phase offset equals the temporal phase offset minus $2\pi\lambda$, where $\lambda$ is wingbeat wavelength. We tested for temporal and spatial synchrony by applying Rayleigh's test for homogeneity of circular data to the temporal and spatial phase delays.

## Estimating wind speed and direction

We estimated local wind speed and direction for each flock using observed variation of ground speeds from birds flying in different directions. Ground speeds and flight directions were calculated for each bird at one-wingbeat time intervals. Median ground speed was calculated for each 10-degree bin having at least 500 data points. A circle was then fit to these median values, with the center of the circle representing a vector of wind direction and magnitude. Ground speeds and wind direction and magnitude were then used to calculate airspeeds. This approach is similar in principle to that used to estimate local wind speed from the drift in the ground reference frame position of circling vultures (*Weinzierl et al., 2016*), and shares the important assumption that airspeed is independent of wind direction. However, birds are theoretically expected and empirically known to vary airspeed with wind speed when flying in order to reach a destination efficiently (*Hedrick et al., 2018*; *Shamoun-Baranes et al., 2007*). Whether this is the case for shorebird flocks (making shorter flocks around the stopover point) is unknown, so we did not attempt to model this possible effect.

We compared our wind estimates to data from nearby weather stations. Our estimated wind direction and speed was typically within ±45 degrees and ±2 m s$^{-1}$ of weather station data (weather station KCAARCAT25). To avoid disturbing the birds, we did not attempt to release helium balloons to measure local wind conditions at altitude. Note that because our analysis here is based almost entirely on the positions and speeds of birds relative to their neighbors, our results are largely insensitive to the wind speed and direction. However, precise determination of bird airspeed and wind direction is required to model the expected position of the wake of the bird, and the absence of this information means that it is not possible to determine when or even if trailing birds interact

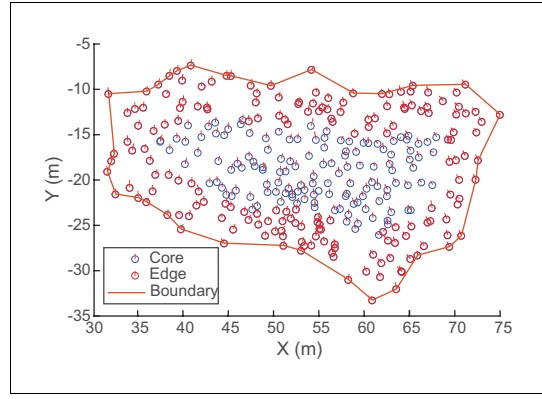

**Figure 9.** Determining flock edge and maximum radius. An overhead view of an example flock of avocets (flock 1220–2). Because flocks were always spread out in the horizontal direction, a compact hull is fitted to the XY-coordinates to create a boundary. The minimum horizontal distance of each bird to the hull is the bird's edge distance. The median edge distance is taken as the flock's maximum radius for computing alignment metrics (*Figure 6*). Here, birds within the maximum edge distance (6.5 wingspans or 4.55 m) are labeled edge, and birds beyond the maximum edge distance are labeled core.

DOI: https://doi.org/10.7554/eLife.45071.026

with the wake of a leading neighbor, or to predict what flapping phase offset would be appropriate for aerodynamically beneficial interaction.

### Statistical analysis

Analyses were conducted using the statistical toolbox in MATLAB r2017b (The Mathworks, Natick, MA, USA). We tested uniformity of circular distributions using Rao's test (*Fisher, 1995*). Because multiple peaks were sometimes present, modal values were calculated using a circular kernel density estimator as an indicator of the predominant alignment direction. For the biomechanical analysis, we used linear mixed-effects models to predict individual wingbeat frequency and airspeed from seven fixed effects—nearest neighbor distance, nearest-neighbor lateral distance, edge distance, airspeed, vertical speed, nearest neighbor species, front-back flock position and hypothesized aerodynamic positioning. Bayesian information criterion (BIC) was used for model selection. All P-values were computed assuming two-tailed distributions.

## Acknowledgements

This work was funded by National Science Foundation grant IOS-1253276 to TLH. We thank Jonathan Rader for assisting with data collection and Jim Usherwood, William Conner and two anonymous referees for providing feedback on earlier drafts of the manuscript.

## Additional information

### Funding

| Funder | Grant reference number | Author |
|---|---|---|
| National Science Foundation | 1253276 | Tyson L Hedrick |

The funders had no role in study design, data collection and interpretation, or the decision to submit the work for publication.

### Author contributions

Aaron J Corcoran, Conceptualization, Software, Investigation, Visualization, Methodology, Writing—original draft, Writing—review and editing; Tyson L Hedrick, Conceptualization, Resources, Data curation, Software, Supervision, Funding acquisition, Visualization, Methodology, Project administration, Writing—review and editing

### Author ORCIDs

Aaron J Corcoran (iD) https://orcid.org/0000-0003-1457-3689
Tyson L Hedrick (iD) https://orcid.org/0000-0002-6573-9602

### Decision letter and Author response

Decision letter https://doi.org/10.7554/eLife.45071.029
Author response https://doi.org/10.7554/eLife.45071.030

## Additional files

### Supplementary files

• Transparent reporting form
DOI: https://doi.org/10.7554/eLife.45071.027

### Data availability

Datasets used in the analysis are included in the manuscript and supporting files. Source data files have been provided for all figures and tables.

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
