## [Decision Letter]

Thank you for submitting your article "Compound-V formations in shorebird flocks" for consideration by *eLife*. Your article has been reviewed by two peer reviewers, and the evaluation has been overseen by a Reviewing Editor and Ian Baldwin as the Senior Editor. The reviewers have opted to remain anonymous.

The reviewers have discussed the reviews with one another and the Reviewing Editor has drafted this decision to help you prepare a revised submission.

Summary:

This paper reports on the dynamics and spatial distribution of individual birds in different flocks of shorebirds. The study covers 4 shorebird species that comprise a seven-fold range in body mass in flocks that ranged from hundreds to a thousand individuals. The flocks were filmed and analyzed using stereo-photogrammetry. The paper reports, for the first time, detailed kinematic and behavioral insight in the formation of shorebird flocks, reporting that on average the horizontal spacing between the birds is about 1 wingspan. Consequently, the spacing is larger between larger birds in absolute distance. Another key find is that the flight speed of individual shorebirds differs depending on the position within the flock. The logical consequence of this find is that these flocks undergo spatial turnover. This behavior may be representative for shorebirds maneuvering as a flock during foraging flights and evading predators, but it is unclear if such dynamics would prevail during long-distance migratory flights, since it may dissolve the flock structure over time. The study thus reports key insights in how shorebirds move as a collective and poses new questions about how this behavior may differ between flight objectives.

Essential revisions:

The reviewers agree with the authors that the basic findings are of interest to a general audience, however, they disagree with the interpretation of the findings and wording in the discussion. The reviewers believe the authors will agree with the reviewers that there may be confounding factors such as the effect of wind gusts and atmospheric turbulence on the displacement and dynamics of vortex wakes that are not fully addressed in the present manuscript. Further, based on state-of-the-art measurement technology it is currently not possible to measure the aerodynamic parameters that would substantiate some of the far-reaching claims made in the manuscript. In case the authors are willing to rewrite the manuscript to remove all un-substantiated discussion (see reviewer comments below) and refocus the manuscript on the valuable and interesting kinematic and behavioral measurements and outcomes, the manuscript could be considered for publication in *eLife*. The revision would include removing most of the discussion and rewriting the key point the authors want to maintain in a condensed format in the very last paragraph: Summarizing it as a speculation on aerodynamic and energetic implications in which the authors propose a need for further study to substantiate these interpretations. All paragraphs before the very last should be speculation free and in accordance with the reviewer comments below. That way a general audience can benefit from the remarkable and interesting findings reported, and it is fine that the precise function cannot be pinned down as of yet, this study will encourage the field to pursue this in follow-up studies. In summary the main concerns are:

1) Crosswind was not established accurately, and this may confound interpretations. We note that it is possible to estimate crosswinds using balloons and tracking, as pioneered by Pennycuick.

2) The flow field generated by birds flying together is unknown and as such any interpretation of spacing in a flock requires speculation. Whereas other papers have speculated on this before, setting a president, we think everyone can agree on that this is a problem (and not a feature) in the literature that we would want to discourage in future papers to advance a more rigorous understanding of collective behavior.

Specific concerns:

1) It is unknown how gusts and atmospheric turbulence may affect the advection and dynamics of wake vortices in the flock, and also the vortex wakes themselves are likely to interact in complex manners that have yet to be quantified. We are concerned that factors are likely confounding the proposed optimal aerodynamic spacing of the birds in the flock. Although it is clear that birds don't need to be benefiting at every instant and that it is probably more reasonable to consider advantages as an ensemble average, care has to be taken to only claim what we can be certain off based on the current data, which do not include flow field measurements. Having shared this reasonable criticism, the reviewers also agree it is reasonable that this could not be quantified at this time or in the near future based on current technology, so we don't expect it to be addressed other than by a manuscript rewrite. This important limitation should be integrated in the discussion and has co-driven our request to move all speculative aspects of the manuscript to the last paragraph of the paper. In this we ask the authors to adhere to the principle "less is more" so they can feature and compel the reader by communicating their main interpretation of the observed kinematics and behavior in the last paragraph while engaging a broad audience with data we all agree on in the main body of the text.

2) The birds would have to fly relatively close to gain noticeable aerodynamic benefit from flying close to the bird in front, but that doesn't appear to always be the case either. This would be worthy of an extra figure so any speculative claims in the last paragraph can be supported, in case these distances are likely too large to give substantial (ensemble) benefits. This data would still be of interest to report, because we currently lack quantitative information on the matters in the literature.

3) The manuscript suggests that the aerodynamic benefit of being in a flock is that individuals fly faster when in the V-optima position. It is unclear to us if this is a benefit, how does that work exactly? How is this in agreement with the objective of individuals in a flock to stick together? Some of the questions that come up for birds flying in a cluster include, where do you speed up to? How does this work spatially and topologically? If birds are using an interaction rule, wouldn't one speeding up cause the others to speed up? Regardless of whether they are in the V-optima position (which is a topological argument)? Is it just a case of individuals of different sizes having to compromise their preferred optimal solo speed to be part of a group? Addressing these issues and questions requires restructuring the manuscript to focus on the data and holding off interpretation and discussion till a concise speculation at the very end of the manuscript. The reason being that the data cannot satisfactory answer these questions, but the data are indeed interesting and thought provoking, which can be highlighted satisfactory by restructuring the manuscript to first present undeniable observations before presenting interpretations that cannot be fully supported by this valuable dataset.

4) In the new speculative last paragraph, we like to see a deeper appreciation of the differences between the new compound-V versus previously reported 'conventional' V formation. Flying in a coherent compound V that remains stable over time requires the individual flight behavior to adhere to topological, spatial, and thus flight dynamic constraints. Questions come up about how flight speed and wingbeat are modulated to fly in a compound-CV? Do the classic adjustment rules for individuals in a flock need to be adjusted (e.g. alignment, speed adjustment, etc.). And how does flying within these constraints enable an individual vs ensemble to save energy? It seems particularly challenging to satisfactory address such questions, and this may also not be needed if the manuscript focusses on reporting the measured outcomes first and foremost, with a reasonable and balanced speculation in the final paragraph encouraging further study of parameters the authors feel will be particularly insightful to support their speculation.

5) The data is insufficient to fully support the putative 'ultimate' benefits of flocking in compound V formations. The proximate factors seem to be better supported and thus more appropriate to highlight.

6) It would be great to clarify the following in the manuscript regarding flight speed and compromise: If the groups don't split up, and they leave and arrive at the same time, what really is the benefit, and how are they gaining an advantage by speeding up exactly?

7) Please rigorously clarify how height was determined in the manuscript and Materials and methods. It takes quite a lot of effort to actually establish this. However, if you are suggesting that birds are in a position to take advantage of upwash to save energy etc., they need to be in the same horizontal plane, or if the wake is advected downward as one expects on basic theory and experiments, they should have some stagger. This needs to be described (albeit briefly) much earlier on and made clear with solid references to the peer reviewed literature. Presently, it's not spelled out and basically not possible to find this information and establish what was done.

8) Please report and clarify the wingbeat phasing in the compound flock. Are we correct to assume that with individual positioning and flaps identified, this can be analyzed? If not, please clarify in the Materials and methods and integrate these limitations in the Discussion.

9) Figure 1 D and G show some movement around the V-optima, what is the relationship between the distance (or delta) between V-optima versus flap frequency? And how does this relate to flight speed? Please report and discuss the relationship between flap frequency and positioning regardless of the effect size. (*eLife* encourages reporting both positive and negative findings).

10) An alternative and equally reasonable hypothesis to the energy hypothesis is that the birds are simply avoiding colliding and end up speeding up and spacing their positions in the flock near-equally accordingly to equally distribute the risk of collision. How does the data support one hypothesis more than the other? The collision avoidance seems more parsimonious at first glance. Please note that *eLife* is interested in the data regardless of which hypothesis it supports best, the dataset and behavioral patterns found are exciting regardless of the precise functional significance, because it is striking and thought provoking in its own right.

11) Please expand the rationale behind the functional tradeoffs/significance for (or against) mixed-species flocks based on the present literature in the Introduction.

12) Please integrate more of the statistics reported in the Materials and methods in the main text (currently the two key results described in the main text are a regression and a Mann-Whitney U test). If there are reasons to believe the current statistical details in the manuscript are sufficient, please explain why in the main text and refer to details in the main text. (Because the current form is somewhat unbalanced and confusing.)

13) The wind speed comparisons between the estimates presented and the weather station are significantly different: a +/- 45° difference in accuracy could have a substantial effect on your findings, please integrate this in your discussion as a limitation. This is one of the reasons we like to see the more speculative discussion/implications in the last paragraph to ensure the overall report is rigorous (except for the last more speculative paragraph).

14) The method of estimating airspeed based on measured ground speed of birds flying in different directions seems troublesome, since it has been established in earlier studies that birds change their airspeed depending on whether they fly in head-, tail- or side winds. Please integrate this literature in your experimental limitations in the Material and methods as well as in the (new) more speculative last paragraph. It is generally accepted that it is more accurate and appropriate to measure local wind directly using anemometer or tracking weather balloons (ascending balloons could easily have been filmed to get wind speed and direction).

15) Whereas we agree the statistics are impressive compared to previous studies, we like to see some caution since the study is based on 18 flocks, of which only two were of mixed species, which is a limited subset of a behavior that remains to be further studied. This further justifies our consensus that speculation needs to be limited to the last paragraph. It would be great if the outlook encourages more detailed study of other flocks across different habitats and periods during the year/migration.

16) Please note and integrate that the "previously-undescribed" flock structure termed "compound V-formation" is a well-known structure among shorebird watchers. Please refer to Piersma et al., 1990, as well as other ecological studies reporting/discussing these behaviors.

17) We do not understand why at one point the authors refer to body size variation among flock member as an explanation, and yet a fixed wing span is assumed for all members of a species in the same analyses. Please clarify the limitations and resolve the confusion for the reader.

---

## [Author Response]

Essential revisions:1) Crosswind was not established accurately, and this may confound interpretations. We note that it is possible to estimate crosswinds using balloons and tracking, as pioneered by Pennycuick.

We agree that the amount error present in our wind measurements may confound our interpretations of the results, and we now point this out in our Discussion. Because of this and the comments below, we have restructured our manuscript to de-emphasize the aerodynamic hypothesis.

2) The flow field generated by birds flying together is unknown and as such any interpretation of spacing in a flock requires speculation. Whereas other papers have speculated on this before, setting a president, we think everyone can agree on that this is a problem (and not a feature) in the literature that we would want to discourage in future papers to advance a more rigorous understanding of collective behavior.

We agree that this is a serious concern and one that cannot be addressed empirically at this time. We have followed closely the recommendations of the reviewers and editors to restructure the manuscript to emphasize our empirical findings and disentangle them from our speculative interpretation that the flock structure might have aerodynamic and energetic benefits.

Specific concerns:1) It is unknown how gusts and atmospheric turbulence may affect the advection and dynamics of wake vortices in the flock, and also the vortex wakes themselves are likely to interact in complex manners that have yet to be quantified. We are concerned that factors are likely confounding the proposed optimal aerodynamic spacing of the birds in the flock. Although it is clear that birds don't need to be benefiting at every instant and that it is probably more reasonable to consider advantages as an ensemble average, care has to be taken to only claim what we can be certain off based on the current data, which do not include flow field measurements. Having shared this reasonable criticism, the reviewers also agree it is reasonable that this could not be quantified at this time or in the near future based on current technology, so we don't expect it to be addressed other than by a manuscript rewrite. This important limitation should be integrated in the discussion and has co-driven our request to move all speculative aspects of the manuscript to the last paragraph of the paper. In this we ask the authors to adhere to the principle "less is more" so they can feature and compel the reader by communicating their main interpretation of the observed kinematics and behavior in the last paragraph while engaging a broad audience with data we all agree on in the main body of the text.

We fully agree with this comment and have restructured the manuscript (and specifically the Discussion) accordingly. More specifically, we have made the following changes:

Abstract: removed the final sentence, which introduced the aerodynamic hypothesis, and replaced it with a statement that we explore multiple potential hypotheses for the observed flock structure

Introduction: changed “to test whether local or global position within the flock affects flight performance.” to “to examine how local and global position within the flock affect flight behavior.”

Discussion: Consolidated the speculative ideas from the last five paragraphs of the original discussion into a single final paragraph, making it clear that the aerodynamic hypothesis remains speculative at this time given the available data..

2) The birds would have to fly relatively close to gain noticeable aerodynamic benefit from flying close to the bird in front, but that doesn't appear to always be the case either. This would be worthy of an extra figure so any speculative claims in the last paragraph can be supported, in case these distances are likely too large to give substantial (ensemble) benefits. This data would still be of interest to report, because we currently lack quantitative information on the matters in the literature.

We agree that birds would need to fly close to one another to interact aerodynamically. As suggested, we have created an additional figure (Figure 3—figure supplement 2) that shows the relative density of nearest neighbors and summarizes the percent of neighbors included in a set of plausible aerodynamic interaction zone. This was 29.74% for the shorebird flocks described here, in contrast to the 4.3% calculated for a chimney swift flock previously measured using a similar videographic approach (Figure 3—figure supplement 1).

3) The manuscript suggests that the aerodynamic benefit of being in a flock is that individuals fly faster when in the V-optima position. It is unclear to us if this is a benefit, how does that work exactly? How is this in agreement with the objective of individuals in a flock to stick together? Some of the questions that come up for birds flying in a cluster include, where do you speed up to? How does this work spatially and topologically? If birds are using an interaction rule, wouldn't one speeding up cause the others to speed up? Regardless of whether they are in the V-optima position (which is a topological argument)? Is it just a case of individuals of different sizes having to compromise their preferred optimal solo speed to be part of a group? Addressing these issues and questions requires restructuring the manuscript to focus on the data and holding off interpretation and discussion till a concise speculation at the very end of the manuscript. The reason being that the data cannot satisfactory answer these questions, but the data are indeed interesting and thought provoking, which can be highlighted satisfactory by restructuring the manuscript to first present undeniable observations before presenting interpretations that cannot be fully supported by this valuable dataset.

We agree that these are important questions, and ones that we are not able to address fully given the available data. Given this, we have decided to highlight these questions as areas for future research in the final paragraph of the Discussion.

4) In the new speculative last paragraph, we like to see a deeper appreciation of the differences between the new compound-V versus previously reported 'conventional' V formation. Flying in a coherent compound V that remains stable over time requires the individual flight behavior to adhere to topological, spatial, and thus flight dynamic constraints. Questions come up about how flight speed and wingbeat are modulated to fly in a compound-CV? Do the classic adjustment rules for individuals in a flock need to be adjusted (e.g. alignment, speed adjustment, etc.). And how does flying within these constraints enable an individual vs ensemble to save energy? It seems particularly challenging to satisfactory address such questions, and this may also not be needed if the manuscript focusses on reporting the measured outcomes first and foremost, with a reasonable and balanced speculation in the final paragraph encouraging further study of parameters the authors feel will be particularly insightful to support their speculation.

As the reviewer points out, our data does not allow us to determine conclusively how and why birds fly in compound-V *vs* simple-V formations. These questions remain for future work, and we now point this out in our final paragraph of the Discussion. Continuing the “less is more” approach suggested by the reviewers, we have decided not to speculate further on these matters.

5) The data is insufficient to fully support the putative 'ultimate' benefits of flocking in compound V formations. The proximate factors seem to be better supported and thus more appropriate to highlight.

We agree with this point and have restructured the manuscript to emphasize our direct findings on the observed proximate factors on bird flocking. We limit all discussion of potential ultimate benefits of flocking to the final paragraph of the Discussion, making it clear that these hypotheses are speculative given the available data.

6) It would be great to clarify the following in the manuscript regarding flight speed and compromise: If the groups don't split up, and they leave and arrive at the same time, what really is the benefit, and how are they gaining an advantage by speeding up exactly?

We believe that this question cannot be addressed with the data at hand and therefore requires a speculative answer, which we have provided in the final paragraph of the Discussion. In brief, 1) the flock may in fact split up, or 2) a greater proportion of the flock may adopt an aerodynamically beneficial alignment during long distance flights as compared to the stopover activity recorded here.

7) Please rigorously clarify how height was determined in the manuscript and Materials and methods. It takes quite a lot of effort to actually establish this. However, if you are suggesting that birds are in a position to take advantage of upwash to save energy etc., they need to be in the same horizontal plane, or if the wake is advected downward as one expects on basic theory and experiments, they should have some stagger. This needs to be described (albeit briefly) much earlier on and made clear with solid references to the peer reviewed literature. Presently, it's not spelled out and basically not possible to find this information and establish what was done.

We have emphasized in the Materials and methods, Results and figures (e.g. Figure 2) that birds were treated as being in the same horizontal plane if they were within +- one wingspan in elevation. It is noted in the first paragraph of the Discussion that on average, this includes 56% of neighbor positions. We have made sure to mention that neighbors are considered to be in the same horizontal plane when they are within +- one wingspan in elevation wherever these results are described. Elevation itself is a direct output from the 3D photogrammetry measurements, with uncertainty similar to the horizontal plane positions. We have elaborated slightly on the Methodological approach to gravity alignment and height determination in the “Three-dimensional calibration”. Finally, because of the complexity and uncertainty of the wake structure, we did not have an a priori prediction regarding the optimal trailing elevation (whether to be level, below or above the leading neighbor). Therefore, we did not include this originally in our results. However, in case this is of interest to some readers in the future, we have added this information to our results (see last sentence of first paragraph in the Results). Finally, a recent study simulating flapping formation flight indicates that the optimal position for trailing neighbors may be above, below, or at the same elevation as the leading bird depending on what is being optimized – see Tay, W. B., Murugaya, K. R., Chan, W. L., and Khoo, B. C. (2019).

8) Please report and clarify the wingbeat phasing in the compound flock. Are we correct to assume that with individual positioning and flaps identified, this can be analyzed? If not, please clarify in the Materials and methods and integrate these limitations in the Discussion.

This information was included in our original submission (see third paragraph of the Biomechanicssection of the Results). We chose not to discuss this finding at length in the Discussion to avoid further speculation regarding potential aerodynamic interactions.

9) Figure 1 D and G show some movement around the V-optima, what is the relationship between the distance (or delta) between V-optima versus flap frequency? And how does this relate to flight speed? Please report and discuss the relationship between flap frequency and positioning regardless of the effect size. (eLife encourages reporting both positive and negative findings).

Position relative to the predicted V-optima had no effect on wingbeat frequency. If one adds the “aerodynamic neighbor” variable to our final model for wingbeat frequency, the Bayesian Information Criterion value increases by 6.7 and this variable is not statistically significant in the model (p = 0.30). We have added this information to the Results.

10) An alternative and equally reasonable hypothesis to the energy hypothesis is that the birds are simply avoiding colliding and end up speeding up and spacing their positions in the flock near-equally accordingly to equally distribute the risk of collision. How does the data support one hypothesis more than the other? The collision avoidance seems more parsimonious at first glance. Please note that eLife is interested in the data regardless of which hypothesis it supports best, the dataset and behavioral patterns found are exciting regardless of the precise functional significance, because it is striking and thought provoking in its own right.

The collision avoidance hypothesis was presented in the original discussion, where we noted that it does not appear to be a good fit to the compound-V formation. We have now expanded this paragraph (Discussion, paragraph four) to more fully explain our reasoning. The main difference between predictions of the collision avoidance hypothesis *vs.* aerodynamics hypothesis is that collision avoidance predicts where *not* to fly, whereas the aerodynamic hypothesis predicts a specific positional arrangement among neighbors. If collision avoidance were the primary factor dictating positioning, we would expect to see much more variation in flock density and modal flock positioning, as is present in other flocking species that have been studied, including starlings and swifts (see newly added Figure 3—figure supplement 1). However, we agree that in the absence of more direct evidence, the ultimate function of the flock arrangements observed cannot be determined.

11) Please expand the rationale behind the functional tradeoffs/significance for (or against) mixed-species flocks based on the present literature in the Introduction.

A review of the benefits and costs of mixed-species flocking is beyond the scope of this manuscript, as this was a relatively small part of what our study addresses. However, we did add a discussion paragraph on the mixed-species flocking result and how our results apply (Discussion paragraph two).

12) Please integrate more of the statistics reported in the Materials and methods in the main text (currently the two key results described in the main text are a regression and a Mann-Whitney U test). If there are reasons to believe the current statistical details in the manuscript are sufficient, please explain why in the main text and refer to details in the main text. (Because the current form is somewhat unbalanced and confusing.)

The statistical tests described in the Materials and methods are Rao’s test for circular distribution uniformity, and a linear mixed-effects model for assessing the relationship of many measurements and properties on flapping frequency and estimated airspeed. The Rao’s test results are in Table 2 and the linear mixed effects model results are presented in Table 3. Because the mixed effect models produce a large list of the individual coefficients and p values we believe they are better presented together in the Table. However, we have added additional references to the statistics tables, and we now provide the associated p-value when specific results are presented in the text.

13) The wind speed comparisons between the estimates presented and the weather station are significantly different: a +/- 45° difference in accuracy could have a substantial effect on your findings, please integrate this in your discussion as a limitation. This is one of the reasons we like to see the more speculative discussion/implications in the last paragraph to ensure the overall report is rigorous (except for the last more speculative paragraph).

We now call more attention to the discrepancy in text (see Estimating wind speed and directionsection in Materials and methods) and point out how uncertainty in the wind speed and direction limits interpretation of other results.

14) The method of estimating airspeed based on measured ground speed of birds flying in different directions seems troublesome, since it has been established in earlier studies that birds change their airspeed depending on whether they fly in head-, tail- or side winds. Please integrate this literature in your experimental limitations in the Materials and methods as well as in the (new) more speculative last paragraph. It is generally accepted that it is more accurate and appropriate to measure local wind directly using anemometer or tracking weather balloons (ascending balloons could easily have been filmed to get wind speed and direction).

We agree that an independent measure of wind speed and direction would be useful and more appropriate for minimizing error. Unfortunately, it was not possible to use balloons at our field site because it would have disturbed many migratory birds, including the birds we were studying, and because it was logistically infeasible to access areas of marsh and bay from where balloons would have needed to be released. We now indicate these limitations in the manuscript (see final Discussion paragraph). We have also expanded the Materials and methods description of our measurement to note the concern that birds may vary airspeed with wind direction.

15) Whereas we agree the statistics are impressive compared to previous studies, we like to see some caution since the study is based on 18 flocks, of which only two were of mixed species, which is a limited subset of a behavior that remains to be further studied. This further justifies our consensus that speculation needs to be limited to the last paragraph. It would be great if the outlook encourages more detailed study of other flocks across different habitats and periods during the year / migration.

We agree with the reviewer’s sentiment. We now conclude our discussion by suggesting further research that emphasizes comparative research on birds flocking in different habitats and conditions.

16) Please note and integrate that the "previously-undescribed" flock structure termed "compound V-formation" is a well-known structure among shorebird watchers. Please refer to Piersma et al., 1990, as well as other ecological studies reporting/discussing these behaviors.

We appreciate the reviewer providing this reference. We have incorporated the descriptions of flocking behavior by Piersma, as well as those of earlier work by Heppner, 1974. We now describe our findings in relation to this research and no longer claim that these are novel flock structures.

17) We do not understand why at one point the authors refer to body size variation among flock member as an explanation, and yet a fixed wing span is assumed for all members of a species in the same analyses. Please clarify the limitations and resolve the confusion for the reader.

The suggestion that the flock segregates by size was put forth as a possible explanation for why birds at different front-to-back positions in the flock had different wingbeat frequencies. We were not able to resolve individual differences in body size (or wingspan), therefore we could not test this hypothesis directly. Because of this, we have removed the speculation of body size influencing flock position as part of the broader effort to include less speculation in the manuscript.